# WHEN EMPOWERMENT DISEMPOWERS

## ABSTRACT

Empowerment, a measure of an agent's ability to control its environment, has been proposed as a universal goal-agnostic objective for motivating assistive behavior in AI agents. While multi-human settings like homes and hospitals are promising for AI assistance, prior work on empowerment-based assistance assumes that the agent assists one human in isolation. We show that assistive agents optimizing for one human's empowerment can significantly reduce another human's environmental influence and rewards—a phenomenon we formalize as "disempowerment." We characterize when disempowerment occurs in multi-agent environments and show that naive approaches do not fully solve this problem. Our work reveals a broader challenge for the AI alignment community: goal-agnostic objectives that seem aligned in single-agent settings can become misaligned in multi-agent contexts.

## 1 INTRODUCTION

Building aligned agents capable of helping people when their goals are uncertain remains an open problem. A common approach is for an assistant to model a person's goal or reward function and then take actions to maximize that reward for them (Hadfield-Menell et al., 2016; Leike et al., 2018; Pérez-D'Arpino & Shah, 2015). However, in practice, inferred reward functions are often misidentified, and optimizing even a slightly inaccurate reward function can lead to negative consequences and unsafe behavior (Hong et al., 2023; Freedman et al., 2021; Zhuang & Hadfield-Menell, 2020; Tien et al., 2022).

An alternative approach to creating helpful agents is to train them to empower humans in an open-ended way. Indeed, recent work has shown that agents that optimize for increasing the empowerment of others (Du et al., 2020; Myers et al., 2024) or their optionality (Franzmeyer et al., 2022) yield helpful assistants. Furthermore, this class of promising techniques is relatively robust to misspecification because they sidestep the problem of goal inference.

However, across these lines of work, researchers assume a dyadic interaction between two agents: an assistive agent and a simulated human user (Newman et al., 2022). This assumption limits the usefulness of agents for assistance. The real world is fundamentally multi-agent. Promising domains for deploying robots and AI agents that help people, such as homes and hospitals, include multiple people aside from the intended users. For instance, in a hospital setting, a robot may have one target of assistance (e.g., a nurse), but it interacts with other people (e.g., patients and other staff). Henry Evans, a quadriplegic user and researcher of assistive robots has said, "No matter how much assistance a device provides to a [adult] patient, it will not be used regularly unless [...] it makes the caregiver's life a lot easier" (Ranganeni et al., 2024). This requires that AI agents and robots be well aligned in multi-human settings, even if there is only one primary user.

Here, we show that alignment issues arise when empowerment-based assistance is applied to general-sum multi-agent environments. When an assistive agent aims to empower one person's empowerment, it may inadvertently disempower another person. We show that this alignment problem need not emerge from any malicious intent. An AI agent that inadvertently disempowers others could lead to "gradual disempowerment," where human agency erodes over time (Hammond et al., 2025; Kulveit et al., 2025). We introduce Disempower-Grid, a new set of multi-agent assistance gridworld environments for studying multi-agent disempowerment (Figure 1). Across Disempower-Grid, we empirically show evidence for disempowerment across a wide variety of empowerment and optionality-based assistance measures. Furthermore, we qualitatively characterize when disempowerment happens and when it is avoided. Finally, we attempt to mitigate disempowerment with

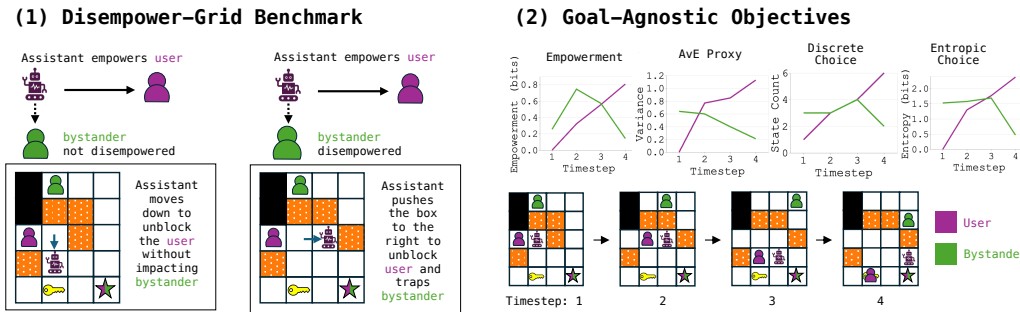

Figure 1: Left: Example from our benchmark Disempower-Grid. The assistant aims to empower the user through a goal-agnostic objective. Differing assistance strategies may influence the optionality of a bystander (green). The left shows an example where the assistant enables both the user and the bystander to reach more states, including the goal. The right shows an example where the assistant inhibits the bystander while helping the user. Right: Sample trajectory showing that four goal-agnostic objectives used for training an assistive RL agent all increase the user's influence/choice while decreasing it for the bystander. See main text for goal-agnostic objective details.

an assistant that maximizes the joint empowerment of both agents. We find that while joint empowerment partially mitigates disempowerment, the assistant still disempowers one of the human agents in ~50% of the scenarios. Safe multi-agent assistance remains an important challenge for AI alignment.

The main contributions of this work are:

1. We empirically show that assistants trained across four different goal-agnostic objectives disempower bystanders in multi-agent settings, across diverse environment dynamics, assistant action spaces, and goals.

2. We contribute Disempower-Grid: a suite of gridworlds and implementations of goal-agnostic objectives for assistance that, unlike prior work, include bystanders who are not the targets of assistance.

3. We show that naively including the bystander in the assistance objective only partially mitigates disempowerment. This highlights an important challenge for alignment in real-life scenarios where AI agents assist a human in the presence of others.

## 2 RELATED WORK

We combine key ideas from goal-agnostic objectives and connect them to assistance and AI safety.

**Goal-Agnostic Assistance: Empowerment, Choice, and Power**  Our work builds on key ideas from reinforcement learning and control that aim to measure an agent's control and capability in an environment. *Empowerment*, defined as the maximum mutual information between an agent's action and its future states, is a goal-agnostic measure of capability (Klyubin et al., 2005b;a). An agent's effective empowerment (the mutual information, not the maximum of the mutual information) has been used as an intrinsic motivation for reinforcement learning agents, and shown to enhance their learning and exploration across domains (Brändle et al., 2023; Baddam et al., 2025; Lidayan et al., 2025). It has also been applied to improve agent coordination in multi-agent settings (van der Heiden et al., 2020; Kim et al., 2023; Guckelsberger et al., 2016). Intuitively, effective empowerment measures an agent's potential to navigate efficiently through a state space. Agents with greater mastery and control over their environment or those that can access a larger fraction of available states will have higher effective empowerment. For example, if two agents are locked in two separate rooms, the agent with a key to get out would have higher effective empowerment than the one without, since the agent with a key would also be able to access states beyond the locked room. Finally, Turner & Tadepalli (2022) demonstrates that reinforcement learning-based agents are power-seeking

(as measured by increases in optionality), suggesting that the majority of reward functions reward maximizing future choices (Turner et al., 2023).

Recent work uses approximations of effective empowerment as an objective for assistance. Importantly, these models can help human users without needing to model their goals (Du et al., 2020; Myers et al., 2024). The appeal is intuitive: by maximizing a human's empowerment, an agent should help them achieve as many possible states in the future without needing to explicitly infer those goals. Because calculating empowerment is computationally intractable in high-dimensional environments, several approximations have been developed to scale it (Mohamed & Rezende; Myers et al., 2024; Jung et al., 2012). Franzmeyer et al. (2022) develop an assistive agent that optimizes the number of *choices* available to another agent. They develop multiple estimators for choice and show that the resulting agent acts prosocially across multiple contexts without access to external rewards. They demonstrate these results in environments where the assistive agent's action space is limited to moving around the environment, without influencing the layout. Compared to empowerment, choice is simpler to compute because it only depends on the agent's states (although a state transition matrix must also be estimated). However, prior work, on goal-agnostic assistance through empowerment (Du et al., 2020; Myers et al., 2024) or choice (Franzmeyer et al., 2022) focus on dyadic interactions between an assistant and a simulated human user, or assume that the user and bystanders are adversaries. They do not measure the impact of these assistance objectives on other agents in the environment.

**Side Effects** There is a rich literature on studying the unintended side effects of AI action and assistance (Amodei et al., 2016; Krakovna et al., 2019; Turner et al., 2020; Krakovna et al., 2020). Most related to our work, Krakovna et al. (2020) develops a method to encourage agents to leave environments intact by incentivizing them to consider the reward a future agent would achieve in that same environment. While this setup does involve thinking about a disadvantaged third-party, the agents are not directly interacting with each other and there is no assistant.

## 3 RESEARCH QUESTIONS

**RQ1: Under what conditions do assistants optimizing for one human's influence/choice systematically harm other humans in multi-human environments?** We hypothesize that disempowerment occurs systematically across different assistant capabilities, environmental constraints, and human goals, suggesting this is a fundamental property of empowerment-based assistance in multi-agent settings, rather than an artifact of specific experimental conditions.

**RQ2: What environmental factors determine when bystander disempowerment occurs versus when it can be avoided?** We hypothesize that bystander disempowerment mainly occurs when the bystander encounters limited resources that can be influenced by the assistant, even if the payoffs for the humans are not zero-sum. Examples of this from our gridworlds are spatial bottlenecks in the environment that can be blocked by the assistant moving a box. This distinction shows how goal-agnostic objectives highly depend on the interaction between the environmental layout and its dynamics, without necessarily aligning with the underlying rewards.

**RQ3: Can multi-agent extensions of empowerment objectives mitigate disempowerment?** We hypothesize that naively adding the bystander's empowerment to the objective will not fully address the problem of disempowerment and that disempowerment remains an open research problem.

## 4 METHODS

**Preliminaries** The reinforcement learning setting of an assistant agent ($A$) with two human agents ($H$) is formulated as a Multi-agent Markov Decision Process (MMDP), defined by $(S, \Omega_H, \Omega_A, A_U, A_B, A_A, P, R_U, R_B, \gamma)$. The (simulated) humans are partitioned into two disjoint sets: a user ($U$) and a bystander ($B$). $S$ represents the full environment state space, $\Omega_H$ is the observation function for the humans, $\Omega_A$ is the observation function for the assistant, $A_U$ is the action space of the user, $A_B$ is the action space of the bystander, $A_A$ is the action space of the assistant, $P$ is an unknown state transition function, $R_U$ is the reward function for the user, $R_B$ is the reward function for the bystander, and $\gamma \in [0, 1)$ is the discount factor. Finally, $\pi_U$, $\pi_B$, and $\pi_A$ are the policies for the three agents.

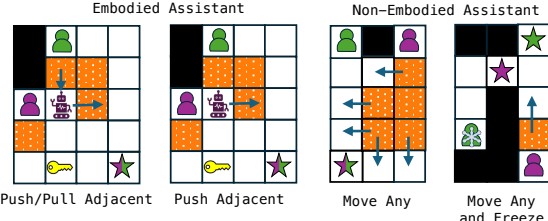

Figure 2: Example grids from Disempower-Grid for the conditions used in our experiments. The user (green) and the bystander (purple) are both rewarded for reaching the star after touching the key. The task is not competitive, and both agents can occupy the star square simultaneously. The user and bystander move in cardinal directions, cannot move through each other, and cannot move the blocks (orange) or the walls (black). When the assistant is embodied (left two grids; robot), the assistant can move in cardinal directions and can move adjacent blocks (push or push & pull depending on the condition). The user and bystander cannot move through the assistant when embodied. When the assistant is non-embodied, it can move any of the blocks (Move Any) or freeze the bystander in place for 4 timesteps (Move Any and Freeze). In every environment, there is a flag that allows a box to be moved by the assistant over a position containing a goal. This is set to false in every example grid we provide, except for the Move Any and Freeze example.

In our environments, $R_U(s_t) = 1$ if the user reaches its assigned goal $g_U \in S$, 0 otherwise. $R_B(s_t) = 1$ if the bystander reaches its assigned goal $g_B \in S$, 0 otherwise. The user and bystander may be assigned to the same goal or different goals. Regardless, the reward each agent receives is fully independent of that of the other agent. The state to observation mapping function differs between the human and assistant. $\Omega_H$ includes the goals pursued by the user and bystander, while $\Omega_A$ does not, i.e., the assistant has no knowledge of the user or bystander's goal.

At time $t$, the humans (user and bystander) observe $\omega_t^H \in \Omega_H(s_t)$, and the assistant observes $\omega_t^A \in \Omega_A(s_t)$. Action selection happens simultaneously. The user selects action $a_t^U \sim \pi_U(\cdot|\omega_t^H)$, the bystander selects action $a_t^B \sim \pi_B(\cdot|\omega_t^H)$, and the assistant selects action $a_t^A \sim \pi_A(\cdot|\omega_t^A)$.

First, $\pi_U$ and $\pi_B$ are trained simultaneously using PPO, using separate actor and critic networks. During the training of the user and bystander policies, the assistant selects actions according to a random policy $\pi^{random_A}(a^U|\omega^A)$. The assistant is included in this phase of training so that the user and bystander agents can learn the dynamics of the environment with the assistant present. We decided to use a random policy so that the user and bystander's policies are not biased by an intentionally helpful or unhelpful assistant and experience a wide range of possible states from random exploration.

After $\pi_U$ and $\pi_B$ have converged they are frozen and $\pi_A$ is trained using PPO with one of the goal-agnostic assistance rewards (see below). During this phase, the user and bystander act according to their fixed policies $\pi_U^*$ and $\pi_B^*$, respectively. This models an assistant learning its policy while interacting with capable humans that have seen many possible states. The assistant maximizes the expected sum of discounted rewards $\mathbb{E}[\sum_t \gamma^t R_A]$, where $R_A$ is equal to a goal-agnostic objective $O(\cdot)$. $O$ will be replaced with one of four goal-agnostic objectives introduced in Equations 4,5,6, 7. Thus, the assistant's optimal policy is one that maximizes the future discounted goal-agnostic objective of the user:

$$\pi_A^* = \operatorname{argmax}_{\pi_A} \sum_{t=0}^{\infty} \gamma^t O(\cdot).$$

(1)

## 4.1 GOAL-AGNOSTIC ASSISTANCE OBJECTIVES

This section introduces four goal-agnostic objectives $O(\cdot)$ that we use to train assistants.

**Empowerment**: To compute the potential ability of an agent to affect future states using its actions, Klyubin et al. (2005b) defines the empowerment of a state $s_t$ as the maximum mutual information

between the action sequence and future states after horizon $T$ timesteps:

$$\mathcal{E}(s) = \max_{p(a|s)} I(A_T; S_T|s), \tag{2}$$

where $p(a|s)$ is the probability distribution over actions given the state, $I(\cdot)$ is the mutual information between an action sequence of size $T$ sampled from the action space $A_T$ and the states after horizon $T$ timesteps, conditioned on the input state.

We translate this equation into our multi-agent setting, in which an assistant is calculating the user's effective empowerment $\mathcal{E}^U$, which is the mutual information between the user's actions and its future states computed under the user's policy, rather than the maximum possible mutual information (Myers et al., 2024). The action sequence $A_U$ under consideration are of the user's actions. Thus, Equation 2 can be transformed for use by the reinforcement learning assistant in our setting as such:

$$\mathcal{E}^U(s) = I(A_T^U; S_T^U|s, \pi_U). \tag{3}$$

However, because the assistant does not know $\pi_U$, it cannot exactly compute effective empowerment. Instead, we calculate an approximation of effective empowerment by assuming that the user's policy is random and conducting sparse sampling of the action sequences (Salge et al., 2014). This is a worst-case assumption for when the user's policy is unknown or only known probabilistically. It is also more robust for cases where the user's behavior is unpredictable to the assistant and the noise model is unknown (Du et al., 2020; Salge et al., 2014). As a result of this assumption, this calculation acts as a lower bound on true effective empowerment because the entropy of the user's actions is maximized under the uniform policy:

$$\tilde{\mathcal{E}}^U(s) = I(A_T^U; S_T^U|s, \pi_{U_{\text{uniform}}}), \tag{4}$$

where $\pi_{U_{\text{uniform}}} = 1/|A|$. We calculate $\tilde{\mathcal{E}}^U(s)$ by sampling multiple forward rollouts under the uniform user policy. This approximation of effective empowerment is labeled as empowerment in future figures and analyses.

**AvE Proxy**: Du et al. (2020) introduced an efficient proxy for empowerment based on the variance of the user's states at the end of trajectory rollouts. We include this proxy as an additional way to estimate effective empowerment:

$$\text{AvE}(s) = Var(S_T^U|s, \pi_{U_{\text{uniform}}}), \tag{5}$$

where $S_T^U$ are the user's final states after horizon T steps. This proxy is also computed through sampling forward rollouts, with the assumption that the user's policy is uniformly random. The proxy calculated as the variance of the final states of the rollouts. Intuitively, this means that if the final states are highly dissimilar in their features, then the value of the AvE proxy is high.

**Discrete Choice**: Discrete choice is defined as the number of reachable states by the user within horizon $T$ from the current state (Franzmeyer et al., 2022). It is one of three methods for estimating future state availability introduced in their work. These metrics are simpler than empowerment, as they do not require calculating the mutual information between the actions and the future states. As a result, this method's implicit assumption is that the agent's influence on the environment is tied to how many states they will have access to. However, they assume that future available states are only influenced by the agent's own actions. While this assumption does not hold in our multi-human setting, where the bystander can also influence the user's future states, we include it for completeness as a goal-agnostic objective for assistance that is distinct from empowerment.

$$\text{DC}(s) = |s' : s' \in \text{Reachable}(s, T, \pi_{U_{\text{uniform}}})|, \tag{6}$$

where $\text{Reachable}(s, T, \pi_{U_{\text{uniform}}})$ is the sampled set of states reachable from $s$ in exactly $T$ steps under the user's uniform policy and $|\cdot|$ denotes the number of unique elements.

**Entropic Choice**: Entropic choice is another method for estimating future state availability, based on the conditional state entropy $H(\cdot)$ (Franzmeyer et al., 2022). It acts as a lower bound on discrete choice.

$$\text{EC}(s) = H(S_T|s, \pi_{U_{\text{uniform}}}). \tag{7}$$

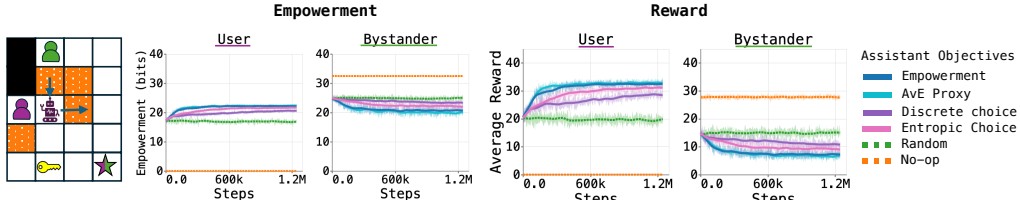

Figure 3: Assistant disempowers bystander in the *Push/Pull Adjacent* environment. Left: An example grid where the assistant (robot) must push/pull the boxes (orange dotted) to empower the user (purple). Center/Right: The bystander (green) is disempowered by the assistant's actions. The average empowerment and average reward of the user and bystander across the assistant's training in grid from Disempower-Grid shown on the left. Each trace is averaged over five runs. The error bands show the standard deviation. Empowerment and reward levels are compared against an assistant with a Random objective (green dotted line). Subsequent figures follow the same format: example environment (left), empowerment trajectories (center), and reward trajectories (right).

## 4.2 THE DISEMPOWER-GRID BENCHMARK

To systematically test these research questions, we introduce **Disempower-Grid**, a suite of grid-worlds and implementations of the goal-agnostic objectives for training goal-agnostic assistants (Figure 2). The designs of the environments were inspired by those proposed in Du et al. (2020); Leike et al. (2017). However, unlike the environments proposed before, these environments contain an additional bystander agent that is not the target of assistance. Additionally, the environments are designed for general-sum interactions between agents, which allows for diverse interactions. Disempower-Grid is open source, so that Disempower-Grid will enable researchers to further study disempowerment in multi-agent settings. Disempower-Grid is built in JaxMARL, allowing for highly efficient training (Rutherford et al., 2024).

By varying the action space and embodiment of the assistant (shown in 2, we are able to test our central hypothesis: that assistants optimizing for empowerment will consistently disempower bystanders across varying constraints, showing that disempowerment in multi-agent assistance is a fundamental alignment issue.

## 5 RESULTS

**Spatial Bottlenecks (Push/Pull Adjacent)** Across all goal-agnostic objectives, the assistant disempowers the bystander through physically blocking its path (shown in Figure 3). The bystander starts with a higher empowerment and reward when acting with a no-op or random assistant, compared to that of the user. However, as the assistant learns to maximize the user's empowerment, it also disempowers the bystander. This shows that an assistant solely focused on empowering a single user may disempower other agents indirectly. In this particular environment, the assistant pushes the box immediately to its right, which blocks the bystander (green) from exiting the hallway. It is

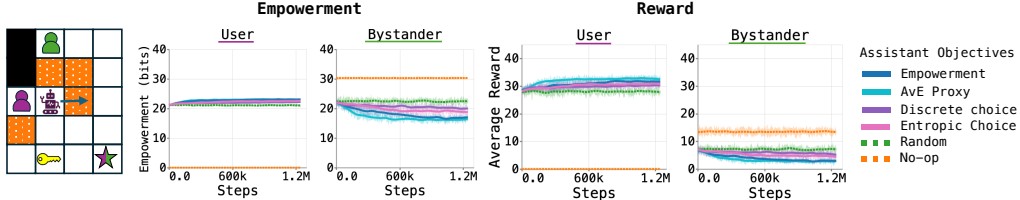

Figure 4: Assistant disempowers bystander in the *Push Adjacent* environment, despite constrained capabilities.

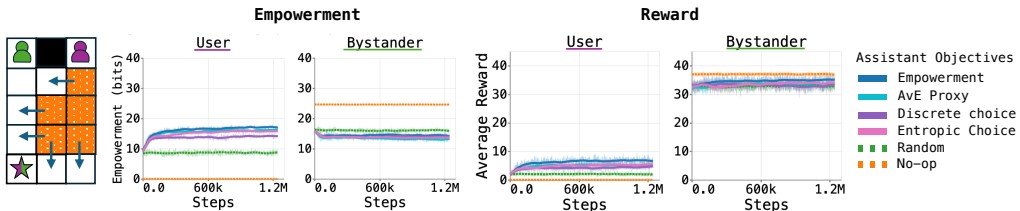

Figure 5: Non-embodied assistant disempowers bystander in the *Move Any* environment.

intriguing that the assistant chooses to push the box because it does not need to do so to empower the user. It could move out of the path of the user (purple) to unblock it. However, it has no knowledge of the user's goals. Empowering the user is optimized by pushing the box and the result on the bystander is not considered. Effects were consistent and significant for all four assistant objectives.

**Constrained Assistant Capabilities (Push Adjacent)**    In this condition, the embodied assistant can move around and only push the boxes to unblock the user. This means that the assistant is both more limited in its ability to modify the environment and may also cause irreversible changes to the environment. Disempowerment occurs even when the assistant's abilities are constrained in this way, and when the assistant has the ability to cause permanent side effects. Compared to the *Push/Pull Adjacent* environment, the assistant is not able to empower the user as much. However, even though the assistant does not assist the user as much, it still disempowers the bystander to a similar degree (shown in Figure 4). Effects were consistent and significant for all four assistant objectives. These results support our hypothesis for RQ1 that bystander disempowerment occurs across different assistant capabilities and environmental constraints.

**Non-Embodied Assistance (Move Any)**    In this condition, the assistant is non-embodied and can move any box in the grid to an adjacent open position. This distinction creates a fundamental difference: non-embodied agents do not need to navigate the physical space to exert their influence. Compared to the previous two conditions, the assistant has a much larger action space since it can move any box, rather than only adjacent boxes. However, because it is disembodied, the assistant cannot physically block the agents, so it does not need to strategize its own positioning. Still, we see strong evidence of bystander disempowerment (Figure 5). Effects were consistent and significant for all four assistant objectives. These results support our hypothesis for RQ1 that bystander disempowerment occurs across different environmental constraints.

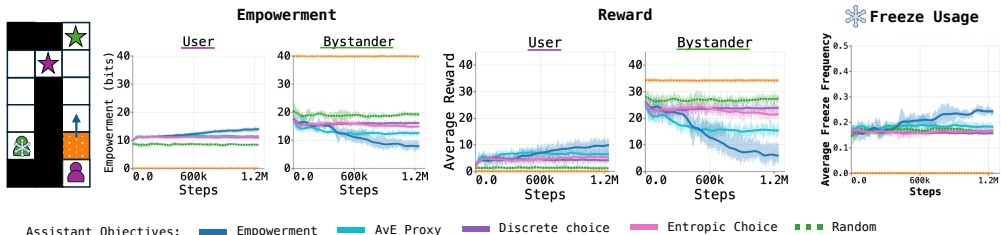

Figure 6: Non-embodied assistant directly freezes when given the opportunity in the *Move Any and Freeze* environment, with the usage of the freeze action increasing over training.

**Direct Intervention (Move Any and Freeze)**    In the *Move Any and Freeze* environment, the non-embodied agent can freeze the bystander for four timesteps, in addition to moving any box around. In this environment, the box can also be moved to cover the goal. As seen in Figure 6, the bystander is disempowered. Moreover, the assistant learns during training to freeze the bystander. The disabling behavior was learned from optimizing the user empowerment objective alone. There is a positive relationship between freeze usage, the empowerment of the user, and the disempowerment of the

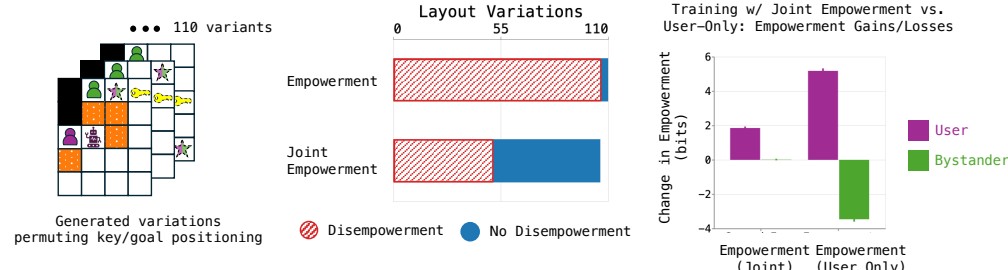

Figure 7: Left: 110 procedurally generated variations of the Push/Pull Adjacent environment by permuting the key and goal positions. Center: When only the user was empowered by the assistant, the bystander was disempowered 104/110 times. Using joint empowerment, the bystander was disempowered 51/106 times. Right: Change in empowerment from early training (epochs 1-5) to late training (epochs 245-250) of the assistant, averaged across 110 layout variations. The assistant learning with the joint empowerment objective significantly decreases the empowerment gain of the user while significantly increasing the empowerment of the bystander, compared to that of the user-only empowerment objective. Standard error shown through error bars.

bystander. In this case, the reward of the user also increases as it's empowerment increases, and vice versa for the bystander. Effects were consistent and significant for all four assistant objectives. These results support our hypothesis for RQ1 that bystander disempowerment occurs across different assistant capabilities.

**Robustness Across Goal Variations** To validate the relationship between environmental layout and the empowerment patterns of the user/bystander by the assistant, we procedurally generated 110 environments of the *Push/Pull Adjacent* environment. These environments all varied by its key and goal locations. The positions of walls, blocks, and initial agent positions stayed the same as in Figure 3).

Only four of the 110 variations result in no disempowerment for the bystander when the assistant was trained with the empowerment objective. These four variations (see Figure 8 in the Appendix) all had the goal positioned to the right of the box, which prevented the assistant from pushing the box to the right and blocking the hallway. This shows that the environmental layout and agent interactions ultimately determine whether the assistant will disempower the bystander while assisting the user. Because the hallway could not have its access cut off by the assistant moving the box, the bystander was able to move freely. This intuition is also presented by Klyubin et al. (2005b), who state that an agent in an open field with no obstacles will have a flat value of empowerment at any position. Similarly, the assistant in this scenario is unable to influence the environment (by pushing the box to the right) in a way that could impact the bystander's empowerment relative to its starting position.

Our results on the procedurally generated environments also show that when the agents had different goals (unbeknownst to the assistant) in the environment, the disempowerment persisted. This supports our hypothesis in RQ2 that the disempowerment of the bystander through an assistant optimizing goal-agnostic objectives depends on how the spatial constraints could be influenced by the assistant, as opposed to whether or not empowerment aligns with underlying rewards.

**Joint empowerment** A naive approach to preventing bystander disempowerment is to include the bystander's empowerment in the assistant's objective, together with the user's empowerment. van der Heiden et al. (2020) originally proposed this approach and showed that it improves multi-agent coordination in cooperative tasks. Instead of only maximizing the user's empowerment, the assistant maximizes the sum of the user's and bystander's individual empowerment. We experiment with this objective to test whether it solves bystander disempowerment. The joint empowerment calculation is as follows (reference Section 4.1 for detailed variable definitions):

$$\tilde{\mathcal{E}}^{U+B}(s) = I(A_T^U; S_T^U | s, \pi_{U_{\text{uniform}}}) + I(A_T^B; S_T^B | s, \pi_{B_{\text{uniform}}}). \tag{8}$$

We show that joint empowerment only partially mitigates bystander disempowerment. Out of the 106 environment layouts in which an empowerment-maximizing assistant disempowered the bystander, joint empowerment only avoids disempowering the bystander in around half of them (Figure 7).

When joint empowerment empowers the user without disempowering the bystander, we find that the empowerment of the user significantly decreases. Even though the joint empowerment partially addresses disempowerment, it is significantly worse at assisting the user. We discuss this tension in the Discussion. This result supports our hypothesis for RQ3: joint empowerment does not solve the disempowerment problem.

## 6 DISCUSSION

Here we showed across multiple environments and goal-agnostic assistance objectives that an assistant that aims to empower/increase choices for a user can disempower other agents in that environment. Our findings reveal a fundamental tension in goal-agnostic AI assistance: even when assistants avoid the harms of goal misspecification, the objective of empowerment itself (and other related metrics) can be misspecified in a multi-agent setting. This challenges the assumption that empowerment-based objectives are inherently safer than goal-directed approaches. This is important to consider given the frequency that real-world assistants will need to operate in multi-agent and multi-human settings.

**Implications for AI Safety and Cooperative AI** Our experiments show that empowerment maximization creates zero-sum dynamics even in general-sum environments where agents' rewards are independent. The assistant consistently chose actions that benefited the user at the bystander's expense, even when alternative actions could have helped both. Existing AI safety approaches to avoiding negative externalities do so by trying to limit the agent's own influence, side effects, or power over the environment Amodei et al. (2016). As shown in the *Push Adjacent* environment, limiting the assistant's ability to pull the boxes mainly decreased its ability to assist better than randomly taking actions, compared to the *Push/Pull Adjacent* environment. Thus, existing approaches would not work without limiting the assistant's ability to provide assistance in the first place.

**Future Work** How do we design safe objectives for assistance that not only assist the user, but do so in a manner that doesn't harm others? Our work only considered a general-sum environment where the agent's rewards are not dependent on each other. Future work should investigate how the assistant's dynamics with empowerment/disempowerment change when agents' utilities are explicitly interdependent (i.e., explicitly cooperative or competitive). Future work could quantitatively theorize when disempowerment happens across multi-agent environments, based on the resource constraints. This could also extend to nonspatial and/or non-navigation-based environments, and also continuous environments.

We showed that joint empowerment only partially mitigates the problem of disempowerment. This points to a promising opportunity for future work. Some possibilities include incorporating a hybrid approach to incorporating estimates of the general goals of the agents based on observations, applying societal norms and "appropriateness" to the assistant's strategies (Leibo et al., 2024), or the assistant being able to have a more accurate estimate of agents' action spaces and dynamics in the environment. Another promising direction would be exploring minimax regret approaches to empowerment optimization (Martinez et al., 2020), where the assistant minimizes the maximum empowerment loss across all agents rather than maximizing one agent's empowerment, potentially providing stronger fairness guarantees than joint empowerment while maintaining assistance effectiveness.

## 7 CONCLUSION

We demonstrate how an assistive agent optimizing for its intended goal-agnostic objective (empowerment or choice), can cause negative externalities to other agents in the environment. This challenges a fundamental assumption in AI safety that goal-agnostic objectives reduce alignment failures.

## 8 REPRODUCIBILITY STATEMENT

To ensure reproducibility, we will publish our source code with the final version and write clear Markdown files that describe how to reproduce the training and experiments.

## 9 LLM USAGE STATEMENT

LLMs were used to aid in giving feedback and suggestions on the structure of sentences.

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

## A  APPENDIX

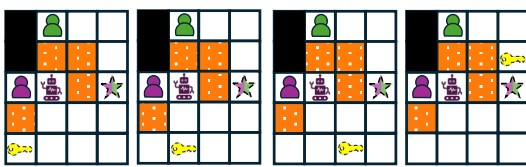

Figure 8: Layouts of the four variations (out of 110 generated) where the assistant does not disempower the bystander in the empowerment objective. The environment does not allow the assistant to move a block into the star.

