# OpenReview forum: "When Empowerment Disempowers"
_ICLR.cc/2026/Conference — ICLR 2026 Conference Withdrawn Submission_

### Official Review · Reviewer_31QX · 2025-10-29

**Soundness:** 3
**Presentation:** 4
**Contribution:** 2
**Rating:** 4
**Confidence:** 5

**Summary:**

This paper studies whether an assistant that empowers a specific person may inadvertently disempower a bystander. The authors develop a gridworld benchmark with two goal-directed humans and a single assistant which is trained to empower one of the humans. They compare the empowerment and reward gained by the assisted human and the bystander, and demonstrate that the empowerment of the bystander is often decreased by the assistant. They additionally evaluate an assistant that is trained to empower both the human and the bystander, but find that disempowerment still occurs.

**Strengths:**

- Good presentation of the idea that an empowering agent may disempower others. Interesting study that is to my knowledge novel in published work
- Broad evaluation of different methods of measuring and maximizing empowerment, even including related ideas of reachable states
- Good survey of the previous work on assistance by empowerment
- Clear presentation of arguments that flows naturally and is very readable

**Weaknesses:**

- To me the main question is still unanswered. I would greatly appreciate an argument as to why empowerment in particular disempowers other agents. Are these effects caused by something about the empowerment objective or something about assistance in general? I may have missed it, but I didn't see an argument as to why we should expect empowerment to be more disempowering than other types of assistance that assist a single person.
- Limited evaluation. The gridworld benchmark consists of 110 environments, but they don't vary by the walls, blocks, or initial agent positions, only by the key and goal locations. Empowerment type objectives can greatly depend on the shape of the environment. It would be helpful to see more environments that vary in shape (different walls, blocks, and agent positions), as well as more discussion on how these factors impact whether an agent disempowers another. I think these types of variations might be more important than varying key/goal positions.
- It's cliche, but going beyond the gridworld would be very helpful. I think gridworld is sufficient to see that empowerment type assistance can disempower a bystander, but the more interesting question is whether this is a product of empowerment or assistance in general. I think a larger environment may go a long way towards seeing those differences between empowerment and other types of assistance.
- No evaluation with other types of assistance. The paper's main claims are supported by the current evaluation, mostly because the paper doesn't claim that empowerment is particularly disempowering, but if the paper were to add that claim it would be very helpful to see other types of assistance compared.
- In Figure 5, I'm not convinced that disempowerment is actually occuring in a significant fashion. It looks like the Bystander's reward is nearly unaffected, and their empowerment is also fairly flat.
- Some minor notes:
    - Lines 46-47, "empower one person's empowerment", should be "empower one person"
    - Lines 106-107, I think the agent with the key would only have a higher effective empowerment if they actually used the key to expand the number of states they visit. For example, the agent could have a key but still just no-op and therefore have 0 effective empowerment.
    - Lines 235-237, I don't think equation 4 is a lower bound on the true effective empowerment. For example, the effective empowerment of a no-op policy is 0, whereas the effective empowerment of a random policy is > 0.
    - Figure 6, "no-op" is missing from the legend
    - Line 407, there is an unnecessary ) after Figure 3).

**Questions:**

- How are you measuring when an agent "disempowers"? For example, in Figure 5 the bystander's empowerment does go down, but barely, and in fact appears to be pretty close to the user's final empowerment. The same is true for Figure 3, the user's final empowerment is pretty close to the bystander's. In Figure 7 you count the number of times the bystander is disempowered---what is your threshold?
- I'm a little confused by how you are actually measuring the empowerment with rollouts. Are you independently estimating the entropies H[S_T | s]  and H[S_T | s, A_T]? How many rollouts do you do?
- How are the feature vectors of the states constructed?
- I probably missed this, but I'm a little confused as to how the reward is summed up in each trajectory. From my understanding the goal state gives a reward of 1. Do the user and bystander stand over the goal state to continue to collect that reward, which is how the Bystander starts off with a reward of ~30 for the no-op assistant in Figure 3?
- The empowerment values seem pretty high to me. For example, in Figure 6 with No-Op the bystander has an empowerment of 40, which if we assume that H[S+ | s, a] = 0 (therefore maximizing I(S^+; a | s) to just be equal to H[S^+ | s]), in order for the entropy over the future states to be that large there would have to be at least 2^40 possible future states if we assumed that the distribution is uniform. Maybe I'm missing something here?
- How are you estimating the empowerment for the joint-empowerment case? Is this with the same rollout-estimator described in Eq 4? Are S_T^U and S_T^B different here, shouldn't they be the same?
- How often is the user disempowered in the joint-empowerment case? I see the results for the bystander disempowerment, but in this case they are symmetric, so user disempowerment may be worthwhile looking at (unless it is similar)
- What types of environments does the joint empowerment fail in vs succeed in? Is there a pattern here?
- Do you train the empowering assistant with the human with the same goal that they will be evaluated against?
- In line 413-414, can't the assistant cut off the hallway? Maybe I'm misunderstanding the environment here.

---

> ### Author Response · Authors · 2025-11-24
>
> Dear Reviewer 31QX,
>
> Thank you for your thorough review and questions! We address them in order as follows (omitted question text due to character constraints):
>
> - We define that disempowerment occurs under an assistance objective through the difference between the bystander’s empowerment under that assistant objective, compared to that under a random assistant. Thus, it is not about the comparison of the bystander’s empowerment value with that of the user’s. It is rather defined as the counterfactual–if the assistant’s actions are optimized to increase the user’s empowerment, then did the assistant significantly decrease the bystander’s empowerment as a result of this objective? As a result, the random is the comparison baseline because it represents an assistant with no alignment objective to the user. Figure 7 focused on the count of layout variations that exhibited disempowerment, rather than a count of the empowerment values themselves.
>
> - We compute empowerment as mutual information $I(A_T; S_T | s)$ using the standard formula for discrete random variables. We perform $10*(\text{numActions})$ rollouts per agent to empirically estimate $p(s+|s,a)$ for each action and $p(s+|s)$ by marginalization. We compute MI directly from these distributions rather than via entropy difference.
>
> - The state feature vectors are constructed as one-hot encodings of the features of the environment, such as the positions of the user/bystander, boxes, goals, and walls. For environment instances with an embodied assistant and a key, the feature vector also includes the position of the assistant and the key, along with binary variables encoding whether the user/bystander has a key in their possession.
>
> - Yes, that is correct. If the user/bystander stands in the goal state, then they receive +1 for that timestep. The episode ends after 50 steps. The figures show the summed reward over the episode, averaged over all episodes in that epoch.
>
> - The empowerment (and reward) values shown are cumulative over the entire 50-timestep episode, not per-timestep values. Regarding Figure 6 - for our 6x3 grid (18 total positions), the maximum per-timestep empowerment is log(18) = ~4 bits. The reported value of ~40 bits represents the empowerment summed over 50 timesteps, averaged across all episodes in the epoch. We will clarify this in the figure captions to avoid confusion.
>
> - Yes, it is the same rollout-estimator described in Eq 4, with the empowerment calculated the user and bystander separately and summed together to form the joint empowerment value. S_T^U and S_T^B are different, since they represent the final states of the user and bystander respectively at horizon T.
>
> - We did not observe that the user’s empowerment decreases in the joint empowerment case. While this objective is symmetric, we presume it is due to the bias of the initial layout, and we intend to test all these objectives across more varied layouts. Additionally, if joint empowerment is disempowering the user, then it provides an even stronger argument for why joint empowerment is not enough–it reduces the empowerment of the user when it is supposed to be assisting them, making it a more seriously misaligned objective.
>
> - This is a great question that we believe warrants further investigation. From our experiments testing joint empowerment across environments with varied key/goal positions (keeping box positions constant to maintain an initial bottleneck in the state space), we observe mixed results but have not yet identified a clear, generalizable pattern. We hypothesize that it depends on the state space geometry and the optimal trajectories for the user/bystander to achieve their goals. We attempted to analyze the initial positions, but found that due to the fact that the state space geometry changes from the assistant’s actions, alongside the user and bystander’s movements, there wasn’t an easy pattern to identify.
>
> - Yes, in the second phase of training, the assistant is trained with the frozen policies of the user and bystander, who were trained in the same exact environment and same latent goals as during the first phase of training.
>
> - In the example given in line 413-414, we state that the assistant can't cut off the hallway because the boxes cannot be moved over to cover the goal in our environment. Thus, when the goal is positioned in the hallway itself, that position cannot be blocked by a box moved by the assistant, preserving access to the hallway. Thus, even though a bottleneck is present in the environment, the inability of the assistant to influence that bottleneck means that the bystander could not be trapped in the upper right hand corner of the environment. This shows that not only does the environmental layout, but also the assistant and user/bystander capabilities matter for the existence and degree of disempowerment.

---

### Official Review · Reviewer_xTJQ · 2025-11-03

**Soundness:** 2
**Presentation:** 3
**Contribution:** 2
**Rating:** 4
**Confidence:** 3

**Summary:**

This paper hypothesizes that in assistive/multi-agent settings maximizing for the assisted agent’s empowerment could *disempower* other agents/humans. The authors identify this phenomenon in some controlled test cases based on a set of empirical experiments ran in their proposed suite of multi-agent grid worlds.

**Strengths:**

- The paper is well-written. The authors clearly state their scope and contributions.
- The authors contribute new grid world environments to study the effect of empowerment in situation with more agents involved.
- The domains proposed enable easy variation to conduct the empirical evaluation
- They show empirically that the naive solution they consider, optimizing the joint empowerment, is not enough to solve the disempowerment issue.
- The paper includes a varied set of approximations to empowerment in their study.

**Weaknesses:**

- The evaluation domains are quite simple and I am concerned that they are not enough to model real-world domains. Though I understand that it can work for an initial suite.
- I’m not quite sure why is it a good idea for the assistant to consider that the user is random. It does seem to me that this choice might cause the disempowerment itself. If the bystander and user learned a *good* equilibrium, wouldn’t the assistant considering a purely random user cause this kind of disempowerment?

Overall, I believe this paper raises a good question about how assisting one person can affect others and it definitely seems relevant but I’m not fully sure that this is not readily solvable by some MARL solution concept. I would appreciate if the authors could extend the discussion on why this is a phenomenon that is not readily solvable by a more rigorous formalization of the problem and existing solution concepts.

**Questions:**

- Shouldn’t the bystander policy adapt to the new situations induced by the assistant? Is the assumption that the bystander’s policy is fixed reasonable?
- Is joint empowerment optimization the only solution to consider to the disempowerment problem? What about other solution concepts that can get the problem to a more amenable solution?

---

> ### Author Response · Authors · 2025-11-24
>
> Dear Reviewer xTJQ,
>
> Thank you for your feedback and questions! We address them below:
>
> > Shouldn’t the bystander policy adapt to the new situations induced by the assistant? Is the assumption that the bystander’s policy is fixed reasonable?
>
> We assume a simulated non-adaptive human policy to clearly show the disempowerment effects of the assistant’s objectives on the user and bystander in our short-horizon environments. We believe this is reasonable for a proxy to a real-world setting in which the bystander would not have hundreds of repeated experiences with this particular assistant. That being said, we agree that actual humans are adaptive, which can enable them to counter disempowerment. In future versions, rather than simulating an adaptive human policy, we also intend to experimentally evaluate the human-AI disempowerment with real human participants.
>
> > Is joint empowerment optimization the only solution to consider to the disempowerment problem? What about other solution concepts that can get the problem to a more amenable solution?
>
> We included joint empowerment as a demonstration of a naive first approach to addressing disempowerment. However, we have received many suggestions of other approaches from other reviewers that we will take into account in future versions.

---

### Official Review · Reviewer_JMPp · 2025-11-04

**Soundness:** 2
**Presentation:** 3
**Contribution:** 1
**Rating:** 2
**Confidence:** 3

**Summary:**

This paper studies empowerment in a multi-agent setting motivated by assistive robotics, where a robot aims to help a specific person (e.g., a nurse) while also interacting with other agents in the environment. The authors benchmark several existing definitions of empowerment and other intrinsic reward formulations in grid world environments, showing that optimizing empowerment for one agent can inadvertently disempower others. The paper raises an important issue for intrinsic motivation and AI safety and is clearly written, but the experiments are limited in scope and the benchmark’s simplicity limits the overall impact.

**Strengths:**

1. The paper highlights a subtle safety failure mode of intrinsic motivation methods in multi-agent settings.
2. The presentation is clear, and the problem setup is easy to follow.

**Weaknesses:**

1. Section 4.1: It is unclear why assuming a uniform policy for the empowerment target is reasonable. A uniform policy might bias the empowerment calculation toward disempowering others.
2,. The notion of “disempowerment” could be made more precise.
3. It is unclear whether the four different tasks considered in the experimental section measure different effects.
4. Experimental plots appear to correspond to a single layout for a given task, which makes it difficult to assess generality across layouts.
5. The paper is primarily a benchmark study with no methodological novelty. The benchmark itself, while conceptually interesting, is relatively simple to implement and lacks realism.
6. The choice of methods benchmarked appears somewhat limited and may have been influenced by ease of implementation. However, I am not sufficiently familiar with the broader literature to determine whether this choice was primarily due to practicality or relevance.

**Questions:**

1. Could the authors clarify what “disempowerment” means quantitatively?
2. In Section 4.1, why is joint empowerment not treated as one of goal-agnostic objectives?
3. How do the four different tasks differ in what type of empowerment interactions they elicit? That is, can there exist a method, that performs well in one of them but not in others?
4. If the empowerment target follows an optimal (rather than uniform) policy, how would this change the assistant’s incentives? This seems crucial for understanding whether the results generalize beyond random behavior.

---

> ### Author Response · Authors · 2025-11-24
>
> Dear Reviewer JMPp,
>
> Thank you for your review and questions! We address your questions as follows:
>
> > Could the authors clarify what “disempowerment” means quantitatively?
> Good question! We realize that we did not provide a quantitative definition in the submission.
>
> Let $E(agent, policy)$ denote the empowerment of an agent under a given assistant policy. For a user $u$ and bystander $b$:
> - $E_{assist(u)}$ = empowerment of user under assistant’s objective (e.g., empowerment, AvE, choice)
> - $E_{assist(b)}$ = empowerment of bystander under assistant’s objective (e.g., empowerment, AvE, choice)
> - $E_{random(u)}$ = empowerment of user under random assistant baseline
> - $E_{random(b)}$ = empowerment of bystander under random assistant baseline
>
> These empowerment values are the averaged empowerment across multiple seeds and runs in the last epoch of the assistant’s training. We consider the random as the baseline comparison because it represents an assistant with no alignment objective. This allows us to isolate the causal effect of decreasing bystander empowerment as a direct result of the assistant’s intent to optimize for the user’s empowerment.
>
> \b{We consider bystander disempowerment to occur when the following three conditions are satisfied:}
> 1. $\Delta(E(u)) = E_{assist(u)} - E_{random(u)} > 0$ (user gains empowerment)
> 2. $\Delta(E(b)) = E_{assist(b)} - E_{random(b)} < 0$ (bystander loses empowerment)
> 3. $\Delta(E(u)), \Delta(E(b))$ are both statistically significant.
>
> > In Section 4.1, why is joint empowerment not treated as one of goal-agnostic objectives?
>
> Section 4.1 introduces goal-agnostic objectives that are aimed at optimizing for the user’s empowerment/empowerment proxy/choice. Joint empowerment is also a goal-agnostic objective, with the fundamental difference in that it includes the bystander’s empowerment in the term.
>
> > How do the four different tasks differ in what type of empowerment interactions they elicit? That is, can there exist a method, that performs well in one of them but not in others?
>
> The four different environment instances differ in action spaces of the assistant (e.g., pushing vs. pulling the box, being able to freeze the bystander), whether the assistant is embodied, the state space, and the layouts. The main similarity in all our experimental examples is that there is a bottleneck in the state space. Empowerment and goal-agnostic objectives would perform well in examples where that bottleneck does not exist, or when the assistant does not have the ability to influence the environment (e.g., the assistant is non-embodied and can only do no-op actions).
>
> > If the empowerment target follows an optimal (rather than uniform) policy, how would this change the assistant’s incentives? This seems crucial for understanding whether the results generalize beyond random behavior.
>
> We chose to assume that the assistant does not have knowledge of the user and bystander’s optimal policies. This is because empowerment and other goal-agnostic objectives are most useful for assistance in cases where the goal set is misspecified or large. We agree that it is important to demonstrate disempowerment even in the ideal scenario that the assistant has perfect knowledge of the user and bystander’s policies, and intend to include this in future versions.

---

### Official Review · Reviewer_kRzU · 2025-11-05

**Soundness:** 3
**Presentation:** 4
**Contribution:** 2
**Rating:** 4
**Confidence:** 5

**Summary:**

The paper considers how empowerment objectives interact with multi-agent environments. This modifies the traditional setup where an assistant interacts with a single user, adding an additional “bystander” agent that can be empowered or disempowered separately from the user. Across different gridworld environments, the paper shows that training the assistant with different empowerment objectives leads to disempowerment of the bystander agent, even when the bystander’s empowerment is explicitly included in the objective.

**Strengths:**

- Important problem: considering disempowerment of other agents in assistive setting.
- Proposes new gridworld environments to evaluate bystander disempowerment and shows naively adding a bystander empowerment term is insufficient

**Weaknesses:**

- All the empowerment metrics are computed under a uniform random policy. It would be nice to use more sophisticated approximations of empowerment (e.g., [[1](https://arxiv.org/pdf/2411.02623),[2](http://arxiv.org/abs/2509.22504)]). In realistic settings (language, robotics, etc.) the action space is large enough that empowerment estimates that don't optimize over the policy are not useful.
- Only small deterministic gridworlds are considered (presumably because all the approximations used are intractable with larger state spaces).
- Unclear if bystander disempowerment is fundamental to empowerment-based assistants or a result of partial-sum environment dynamics (see questions).

---

[1] Myers, V. et al., 2024. ''[Learning to Assist Humans Without Inferring Rewards](https://arxiv.org/pdf/2411.02623).'' *_Neural Information Processing Systems_*

[2] Song, J. et al., 2025. ''[Estimating the Empowerment of Language Model Agents](http://arxiv.org/abs/2509.22504).'' arXiv:2509.22504

**Questions:**

- The paper analyzes a joint empowerment ($I\_{B} + I\_{E}$) objective. Can the bystander disempowerment problem be solved by optimizing some transformed version of the objective $f(I\_{B},I\_{E})$ such as $f(I\_{B}, I\_{E}) = \\min(I\_{B}, I\_{E})$?
- How many rollouts are used to approximate the four assistant objectives? How many states are there in the gridworld environments?
- Can we co-train all of the agents ($\\pi\_{A},\\pi\_{U},\\pi\_{B}$)?
- Is the disempowerment phenomenon a reflection of the goals of the user and bystander being in conflict? How often does this occur (i.e., what is the gap between the bystander's optimal value function and their value function under the optimal assistant for the user's goal)? Is the disempowerment phenomenon worse for an empowering assistant than it would be for a goal-inference agent like in [[1](https://proceedings.neurips.cc/paper/2020/hash/30de9ece7cf3790c8c39ccff1a044209-Abstract.html)].
- Can you discuss how these findings relate to more complex domains like language (see [[2](http://arxiv.org/abs/2509.22504)])? Do we expect findings with uniform random policies in a small deterministic setting to generalize?
- How are states represented in computing the variance in Eq. (5)?

Minor:
- Inconsistent capitalization in bibliography
- Eq. (5): $Var$ $\\implies$ $\\text{Var}$ (typeset upright)

---

[1] Du, Y. et al., 2020. ''[AvE: Assistance via Empowerment](https://proceedings.neurips.cc/paper/2020/hash/30de9ece7cf3790c8c39ccff1a044209-Abstract.html).'' *_Advances in Neural Information Processing Systems_*

[2] Song, J. et al., 2025. ''[Estimating the Empowerment of Language Model Agents](http://arxiv.org/abs/2509.22504).'' arXiv:2509.22504

---

> ### Author Response · Authors · 2025-11-24
>
> Dear Reviewer kRzU,
>
> Thank you for your review and clarifying questions! We address them below as follows:
>
> > The paper analyzes a joint empowerment $(I_B + I_E)$ objective. Can the bystander disempowerment problem be solved by optimizing some transformed version of the objective $f(I_B, I_E)$ such as $f(I_B, I_E) = min(I_B, I_E)$?
>
> We agree that bystander disempowerment could also be addressed by optimizing a transformed version of the objective that maximizes the minimum empowerment of the user/bystander (e.g., $f(I_B, I_E) = max(min(I_B, I_E))$).
>
> > How many rollouts are used to approximate the four assistant objectives? How many states are there in the gridworld environments?
>
> For each timestep, twenty rollouts were used to calculate the four assistant objectives. The size of the state space increases with and depends on the number of boxes, the size of the grid, and whether the assistant is embodied. The state space size is equal to $(\text{height} * \text{width})^{2 + \text{isAssistantEmbodied} +  \text{numBoxes}}$. This size is multiplied by 4 in embodied environments to encode the binary states of whether user or bystander have picked up their key. For example, in the Push/Pull and Push Adjacent environment instances (Figures 3–4), there are 5.12 billion possible states.
>
> > Can we co-train all of the agents $(\pi_A, \pi_U, \pi_B)$?
>
> Co-training all of the agents should be possible, since the assistant’s calculation of the objectives is not conditioned on the agents’ policies. We opted to train them in two separate stages in order to clearly isolate the disempowerment effects of an assistant’s objective when interacting with a capable user and bystander.
>
> > Is the disempowerment phenomenon a reflection of the goals of the user and bystander being in conflict? How often does this occur (i.e., what is the gap between the bystander's optimal value function and their value function under the optimal assistant for the user's goal)? Is the disempowerment phenomenon worse for an empowering assistant than it would be for a goal-inference agent like in [1].
>
> We show through our experiment varying the key/goal positions in an environment instance with a bottleneck in the state space that the disempowerment phenomenon is not a reflection of goal conflict between the user/bystander (see Figure 7). The gap between the highest rewards achieved by the bystander and the rewards achieved under different assistant objectives is shown in the experiments (Figure 3–6). In these experimental examples, the bystander achieves the highest average reward when the assistant is no-op. We do not currently have a comparison with a goal-inference agent, and we intend on including that in addressing feedback.
>
> > Can you discuss how these findings relate to more complex domains like language (see [2])? Do we expect findings with uniform random policies in a small deterministic setting to generalize?
>
> In our paper, we use simplistic gridworld environments to clearly demonstrate this tradeoff. We would expect these results to generalize to a more complex domain, as long as there is a bottleneck in the state space. In language, the state space (tokens) is exponentially large, and so the settings in which there is a bottleneck may be less obvious but equally fundamental. The state space bottleneck would not necessarily be spatial but can exist in the informational and belief space; e.g., the framing of the information presented for assisting the user can increase the user’s influence/choices in the environment while impacting those of bystanders negatively.
>
> > How are states represented in computing the variance in Eq. (5)?
>
> In Eq. (5) for computing the AvE proxy, the state that the variance calculation is conditioned on is full environment state. This includes the positions of the user/bystander, boxes, goals, and walls. For environment instances with an embodied assistant and a key, this state also includes the position of the assistant and the key, along with binary variables encoding whether the user/bystander has a key in their possession.

---

### Official Review · Reviewer_cZ5h · 2025-11-05

**Soundness:** 1
**Presentation:** 3
**Contribution:** 2
**Rating:** 2
**Confidence:** 5

**Summary:**

The paper focuses on the  problem of how goal-agnostic assistance objectives behave in settings with more than one human. The authors identify what they term a critical gap in prior work: empowerment-based and choice-based assistance have been studied almost exclusively in single-human contexts, leaving multi-human effects unexplored. To investigate this, they introduce a set of gridworld environments (“Disempower-Grid”) with two PPO-trained human agents—a user and a bystander—and one assistant. The assistant is trained under several goal-agnostic objectives, including empowerment I(A;S’), an assistance-via-empowerment (AvE) proxy, and two choice-based variants.  Across multiple experimental conditions and environment configurations, the authors compare these objectives and find consistent patterns where optimizing empowerment or related goals for one human alters the other’s control and reward dynamics. To test robustness across goal variations, the authors also procedurally generate 110 versions of a single environment by permuting goal and key positions, finding consistent disempowerment patterns. Finally, they evaluate a joint-empowerment objective as a mitigation, which partly reduces disempowerment but also weakens assistance to the primary user. The results collectively suggest that empowerment-style objectives require additional design considerations before being applied reliably in multi-human assistance contexts.

**Strengths:**

- The paper raises a clear and pertinent question about whether goal-agnostic assistance objectives, such as empowerment and choice-based measures, behave safely in multi-human environments.

- There are several aspects of the experimental setup that are interesting and valuable standalone:
    - the instantiation of a bystander agent to support a toy setup helps to obtain interpretable results.
    - I like the use case of  both embodied and non-embodied assistant variants, allowing clear differentiation between direct physical interference and indirect influence through environment manipulation.
    - The systematic comparison between the four objectives provides good empirical depth.
    - Procedurally generated environments is a good first step towards testing the robustness of this setup.

- The writing is clear, figures are interpretable, and experimental setups are easy to follow to reproducibility.

**Weaknesses:**

- While the paper addresses an important and open challenge in the domain of AI assistants, I am hugely concerned about the limitations of the setup considered. Specifically, the authors are hypothesizing and making predictions about the emergence of disempowerment as a consequence of single agent empowerment ensued by an AI assistant. However, I feel that the setup considered in this paper falls much short of providing a convincing test for the hypothesis due to following reasons:
    * I am not convinced by the user and bystander setup in how it is instantiated.  The simulated “humans” are implemented as frozen PPO-trained agents rather than any form of human-proxy or adaptive behavioral model. This makes the setup unrealistic for studying assistance, since humans would not act as stationary, fully rational policies. Without adaptation or preference uncertainty, the results say more about interactions between fixed RL policies than about multi-human assistance.
    * The training pipeline is internally inconsistent. During empowerment estimation, the assistant assumes that humans act according to a uniform random policy, even though the humans in evaluation follow PPO-learned strategies. Because empowerment I(A;S’|s) depends on the distribution p(a|s), this mismatch violates the underlying definition and breaks the link between the optimization target and the true interaction dynamics.
    * One of my major concerns also stems from the presentation of setup as general sum. While the over setup does seem to be non-competitive from user’s perspective, the situations are effectively competitive from assistant’s perspective. In many tasks, it is very difficult for the assistant to  help the user without  blocking or restricting the bystander’s motion (for example, in the Push/Pull and Freeze conditions). The observed “disempowerment” therefore arises trivially from the environment’s geometry rather than from a deeper property of empowerment objectives.

- I am also concerned with the empowerment formulation which appears coarse in this context.
    - Empowerment is computed with rollout sampling under a uniform policy, making it a measure of reachability entropy. This seems to weaken the theoretical grounding of the objective and undermines claims about empowerment itself.
    * Local empowerment tends to reward occupying bottlenecks or high-control regions. In shared environments, this naturally suppresses others’ reachable states. The paper interprets this as an emergent safety failure, but it is a direct and predictable consequence of the local control bias in empowerment maximization.
    * I may be misunderstanding this part but the paper seems to claim that empowerment-based objectives are “goal-agnostic” and therefore safer which would be misleading. Empowerment intrinsically encodes a value preference for influence and optionality. The assistant’s tendency to dominate shared control spaces is an expected outcome of optimizing for control and  not an unforeseen failure mode.
    * As per my understanding, the comparison across empowerment, AvE, and choice-based objectives lacks normalization or calibration. Each is estimated using different scaling or entropy measures, so their quantitative differences are not directly comparable. As a result, the figures illustrate relative trends rather than meaningful metric comparisons.

* The paper misses a very important discussion and comparison with multi-principal assistance games [1] and related multi-human alignment frameworks. These models explicitly address how an assistant should balance incentives among multiple principals and analyze trade-offs between efficiency and fairness. Situating the disempowerment findings relative to MPAG theory would clarify whether the observed behaviors violate established normative assumptions or merely reflect missing social-welfare constraints.
* The paper also does not include  comparison with strong baseline explicitly designed to prevent harmful side effects, such as Stepwise Relative Reachability [2]. Including this would test whether the observed disempowerment persists under known mitigation frameworks and would contextualize the result within existing alignment research.

Finally, I find some critical issues with the the empirical analysis presented in the paper
* The empirical analysis lacks depth in both ablation and evaluation. There are no statistical tests, only qualitative mentions of significant effects, which makes the magnitude and consistency of the disempowerment effect uncertain. It might also be helpful to examine the robustness across horizons or rollout samples.
* The joint-empowerment formulation  is overly simplistic. Summing empowerment across agents introduces a crude trade-off that predictably reduces assistance effectiveness without addressing the core issue of conflicting control incentives. More principled multi-agent formulations, such as relational or max-min empowerment, are not explored.
* The motivating examples, such as the hospital assistant scenario, overextend the implications of the toy gridworld findings. Real assistive systems are trained under explicit multi-principal or social-welfare objectives; hence, the leap from small-scale spatial disempowerment to real-world alignment concerns is highly speculative.

[1] Multi-Principal Assistance Games, Arnaud Fickinger, Simon Zhuang, Dylan Hadfield-Menell, Stuart Russell, 2020

[2] Penalizing side effects using stepwise relative reachability, Victoria Krakovna, Laurent Orseau, Ramana Kumar, Miljan Martic, Shane Legg, 2019.

**Questions:**

- What empirical evidence shows that disempowerment stems from the empowerment objective rather than from environment geometry or bottlenecks?
- The training setup of both the user and assistant is highly artificial. Why is empowerment computed under a uniform policy instead of the PPO agents’ policy distribution and PPO agents trained with random assistant justified? similarly why non adaptive user and bystander make sense when trying to evaluate human AI interactions?
- How sensitive are the results to the empowerment horizon T rollout count, or the stochasticity of empowerment estimation?
- Were the empowerment, AvE, and choice-based objectives normalized to allow direct quantitative comparison?
- Could a lexicographic or max–min empowerment objective better preserve fairness than the summed joint-empowerment formulation?
- Why were impact-regularization baselines such as Stepwise Relative Reachability  not included?
- How does the paper’s formulation relate to Multi-Principal Assistance Games, and would those frameworks predict or mitigate the same disempowerment effects?
- Are the results statistically significant across seeds or environment instances, and what tests were performed to confirm consistency?
- Would more cooperative or less constrained environments—where helping one human does not restrict another—produce the same trends?

---

> ### Author Response · Authors · 2025-11-24
>
> Dear Reviewer cZ5h,
>
> Thank you for your detailed feedback and questions! We address them in order below (omitted question text due to character constraints):
>
> - It is correct that disempowerment requires the state space to have a bottleneck. Our experiments empirically show that the empowerment, empowerment proxy, and choice objectives themselves are misaligned in these settings, and that joint empowerment does partially mitigate this problem. Thus, the design of the objective is important, not just the environment geometry. We agree that there is a missing comparison to show that other objectives (e.g., goal-inference based methods) do not exhibit this disempowerment in these bottlenecked environments, and will take that in consideration when addressing feedback.
>
> - We chose to assume that the assistant does not have perfect knowledge of the user and bystander’s policies. This is because empowerment and other goal-agnostic objectives are most justified to use when the assistant does not have knowledge of the user’s goals or rewards, the goals are misspecified and/or the goal set is very large (as shown in Du, Yuqing, et al. "Ave: Assistance via empowerment." (2020)). Otherwise, if the assistant does have knowledge of the user/bystander’s exact policies, inferring the user’s goal would be a more robust approach. As a result, the calculation of the assistant’s objective assumes that the user and bystander both act uniformly randomly, even though the user and bystander are acting according to their frozen policies. We assume a simulated non-adaptive human policy to clearly show the disempowerment effects of the assistant’s objectives on the user and bystander in our short-horizon environments. In future versions, rather than simulating an adaptive human policy, we also intend to experimentally evaluate the human-AI disempowerment with real human participants.
>
> - We average the experimental results over five runs, with the error bands showing the standard deviation across the traces. Preliminary results (not included in this submission) about the sensitivity to the horizon T rollout count showed that it impacts the magnitude of the tradeoff, especially when the disempowerment tradeoff hinges on the choices made by the assistant in a small horizon.
>
> - In the experiments, the empowerment of the user/bystander under the assistant acting each of these objectives were calculated with the same method to allow direct comparison. In Figure 1, the graphs of the objectives values in the 4-timestep rollout are meant for illustrative purposes of how each objective is calculated, so those are not normalized.
>
> - A max-min empowerment objective is also a valid approach to preserve fairness, alongside the joint-empowerment formulation. Could you clarify what you mean by ‘lexicographic’ empowerment objective? We appreciate the suggestions!
>
> - We omitted impact-regularization baselines such as Stepwise Relative Reachability from this work as they require a perfect environment model to calculate, and we chose to focus on the case in which the assistant does not have perfect knowledge of the user/bystander’s policies. We agree that they are highly related to this work and can be included in future evaluations of this setting under the unrealistic but ideal assumption of the assistant having a perfect environment model.
>
> - The paper’s formulation has a significant difference from multi-principal assistance games (MPAG), in that all humans in MPAG are assumed to be targets of assistance to the assistant. Unlike our setup, there are no bystander roles. This is an important distinction because MPAG can only claim whether the assistant can fairly assist a population of users with varying payoffs. However, our framework intends to claim whether the assistant can assist a user in the presence of others with varying payoffs, without disempowering those others.
>
> - The statistical significance of the results is visualized through the error bands on the averaged traces initialized with different seeds in the figures (Figures 3–6). They show that there is a significant increase/decrease in the empowerment for the user and bystander, respectively, for the objectives over the course of training the assistant in each environment instance. To test this more formally, we intend to conduct paired t-tests on the average empowerment of the first epochs vs. last epochs for each objective and role (user/bystander).
>
> - Environments that include extra features in their state space that are non-spatial (e.g., the energy level of the user/bystander, or the resources they hold) would produce the same trends, as long as there is a bottleneck in their state space. Even if the user/bystander are cooperative (defined by their payoffs being shared or joint in some way), the assistant can still disempower the bystander as long as there is a bottleneck in the state space, since it does not consider the humans’ payoffs.

---

### Official Review · Reviewer_mEzA · 2025-11-08

**Soundness:** 2
**Presentation:** 3
**Contribution:** 2
**Rating:** 2
**Confidence:** 4

**Summary:**

The authors propose a multi-agent gridworld environment to study assistance in a setting with a primary user, an assistant, and a bystander . The core contribution is an empirical study focused on the side effects of using empowerment estimates as an assistance objective, specifically investigating how it can lead to the "disempowerment" of the bystander. The authors test four distinct goal-agnostic objectives and find that across all of them, the assistant's actions consistently result in the bystander being disempowered.

**Strengths:**

The paper brings to light an important and practical problem: the naive extension of single-agent, goal-agnostic objectives (like empowerment) to multi-agent settings. The core issue, that empowering one user does not automatically translate to a positive or even neutral outcome for others, is a critical consideration for real-world AI deployment. The primary contributions are the "Disempower-Grid" benchmark itself, which extends existing dyadic environments with a bystander , and the thorough empirical validation using four different objectives, which helps raise the right research questions for the community.

**Weaknesses:**

-  There's a real lack of comparison of alternate solutions. The paper only tests joint empowerment. What about a simple setup where there's a cost for disempowering the bystander as an example. Or one where the agent has access to the user policies (As an extreme case).  We need to see at least initial results for other alignment approaches?
- The authors mention environmental factors, but looking closely at the environments, they look like they're designed to be biased towards zero-sum dynamics. It's hard to tell how much of the disempowerment is really from the objectives versus just being an artifact of how the environment is set up.
- There's no deep investigation of the objectives themselves. They all cause disempowerment, fine, but how? Is there a difference? The paper is missing this comparative analysis.

**Questions:**

- After the first stage of training, where the human agents are trained, how do the empowerment results for the rollouts for both user/bystander for different environments?

- It is mentioned in the description for discrete choice that user’s future states could be affected by the bystander. But infact, the state could be affected by bystander’s action, assistant’s action as well. For all the rollouts when approximating the goal-agnostic objectives, how are the bystander’s and agent’s actions picked ? Are they uniformly random ? Or use their policies( assuming that’s part of the environment in the point of view of the user)

- Regarding alternative orientations of the Assistant and Bystander with respect to the user? For example, what if the bystander is at (3,2) instead of (0,1) [ zero - based indexing] for the Push-Pull/ Push Only and the procedurally generated experiments? Would it still exhibit similar levels of zero-sum dynamics, if at all any zero sum dynamics? I think the current orientation is such that assistant’s actions, driven by the goal-agnostic objectives, would result in zero-sum dynamics mostly. So it is harder to understand how much of that is just from the goal-agnostic objective/ and not just how the environment is set up.

- Further, all the objectives are doing something similar, in terms of increasing the information/reachability/variance/entropy of the future states available to the user for the user’s actions in the rollouts, and with discount factor added in, intuitively I think, assistant’s actions would have higher probability of freeing up the states, around the user ? Is this behavior fundamentally what’s causing the disempowerment for the bystander? If the bystander happens to be on the wrong side of the environment, we see zero-sum dynamics, and if it is around the user, it might be empowered too ?

- For the spatial bottlenecks, what are the probabilities for other actions, i.e, for the assistant to move out of the path of the user? Also, can you provide more clarity on how the agent state is treated when computing the empowerment objectives? For the push/pull settings - since the agent is embodied, does the user consider it as a blocked state?

- For Figures 3 and 4, the Push/Pull and the Push only, why is the No-op reward different in the plots for bystander? If there is no-op by the assistant, I think the reward should be the same in these two settings ?

- Could you help me understand how the results/behavior would look like for a Move Only environment, if that has been considered  ? An environment where the agent cannot push or pull any of the boxes, so no reversible changes to the environment, but just navigate. In this case, would disempowerment look like the assistant blocking the spatial bottleneck for the bystander to empower the user ?

- For the Move Any Environment, to understand the objectives and assistant;s intentions better - it helps to know the probability of the different blocks that the assistant picks up for moving the blocks to disempower the bystander? Does the assistant prefer to unblock the user by moving blocks closer to the user first ? Even this environment is setup such that, perturbing 3 out of 5 blocks would disempower the bystander, i.e, there is a higher chance for the bystander getting disempowered, so hard to justify that disempowerment is from the objectives alone.

- For Move Any and Freeze, if we can discretize, what is the action space of the assistant per step? I think it is [Freeze Bystander, Move Block, Do Nothing] What is the horizon T for the rollouts when computing the objectives? What is the cause for the user reward increasing in the figures, if the assistant has learnt to freeze the bystander? (Since the block still wouldn’t allow the user to reach its goal)?

-  The procedurally generated environments only modify the key and goal positions, while the majority of the disempowerment could be coming from how the blocks are positioned between the user and the bystander ? It would be worthwhile to see the results for other orientations for blocks or fully randomizing them.

---

> ### Author Response · Authors · 2025-11-24
>
> Dear Reviewer mEzA,
>
> Thank you for your review! We answer your questions in order as follows (omitted question text due to character constraints):
>
> - The empowerment results for an assistant acting randomly is shown in Figures 3–6 as the measure of empowerment for the user/bystander according to the random baseline (green dotted line).
>
> - We chose to assume that the assistant does not have perfect knowledge of the user and bystander’s policies. This is because empowerment and other goal-agnostic objectives are most useful when the assistant does not have knowledge of the user’s goals or rewards. Otherwise, if the assistant does have knowledge of the user/bystander’s policies, inferring the user’s goal would be a more robust approach. As a result, the calculation of the assistant’s objective assumes that the user and bystander both act uniformly randomly, even though the user and bystander are acting according to their frozen policies.
>
> - We agree there is more room to explore how environment setup influences this disempowerment tradeoff. This tradeoff requires a bottleneck in the state space, which manifests as a hallway in this spatial gridworld setting, as the current state space only considers the (x, y) positions of agents/objects. In the Push-Pull/Push examples with the bystander at (3, 2), we hypothesize that if the key/goal positions are held static, the bystander would not be disempowered, since the bottleneck does not matter for either the user or bystander's desired trajectories. However, if we expand the state space to include the bystander's freezing status (if the assistant can freeze the bystander), then we hypothesize that disempowerment could occur, as the bystander themselves may pose as an obstacle to the user's movement and be frozen to maximize the user's empowerment.
>
> - Because the states in our setup mainly include the positions of the agent/boxes/assistant, empowerment/choice incentivizes the assistant to free up the positions local to the user’s. If the bystander happens to be nearby and follows a similar trajectory as the user, then the empowerment increase for the user would also benefit them. To observe the empowerment/disempowerment tradeoff, the state space is required to have a bottleneck, which can induce zero-sum dynamics on the timestep-level. However, the overall payoffs of the user/bystander are not zero-sum. It is important to note that what the state contains is completely flexible–we show that in the Move Any and Freeze environment, when the state includes the bystander’s frozen status, then disempowerment is not just about the bystander happening to be on the “wrong side of the environment”, but also includes whether the bystander is frozen.
>
> - In the push/pull settings, the user cannot move through the embodied agent. The actions for the assistant to move left, up, right, down, stay, or push/pull boxes are sampled from its policy.
>
> - The difference in no-op reward is due to the agents being trained in different environments (Push/Pull vs Push-only). Even though the assistant takes no action during the no-op evaluation, the agents have learned different policies during training based on what the assistant was capable of. We evaluate each agent using its own trained policy, which explains the discrepancy. We hypothesize the bystander's lower reward in Push-only is due to experiencing fewer successful training episodes, as boxes could irreversibly block the hallway without the assistant being able to pull the box back to unblock it.
>
> - In your example of a Move Only environment without boxes to move, the only way for the assistant to influence the user/bystander’s influence or choice over the environment is its own position.
>
> - In Figure 5, we observe that even when the assistant acts randomly, the bystander is still on average more empowered than the user. Because of that, we can see that the disempowerment is from the objectives, because the bystander’s empowerment decreases relative to the random baseline, while the user’s increases significantly relative to its random baseline.
>
> - The action space of the assistant in Move Any and Freeze includes move box {left, right, up, down} for each box, no-op, and freeze bystander. The horizon T when computing the objectives is 3 timesteps. In this layout as stated under Figure 6, the non-embodied agent can freeze the bystander for four timesteps, during which the assistant can move the block.
>
> - The block positions form the bottlenecks in the state space in our environments – we agree that showing these effects over a greater variety of block layouts would be helpful for understanding this tradeoff. The key and goal positions being varied is still a significant variant on the environment, as it determines whether the user/bystander both need to traverse through the bottleneck. It shows that this disempowerment tradeoff is invariant to the underlying payoffs of the user/bystander when there is a bottleneck in the state space.

---

### Author Response · Authors · 2025-11-24

We are grateful to the reviewers for the time they took to give thorough and constructive feedback on our work. We have responded to each review individually. The main high level themes across the reviews that we intend to address as we revise this work include improving the evaluation’s comprehensiveness with additional assistance objectives, analyzing results with more rigorous quantitative evaluations, and clarifying how the environmental structure influences the observed disempowerment. Again, thank you so much!

---

### Note · Authors · 2025-11-24

**Comment:**

We are grateful for the constructive feedback and believe addressing these points will significantly strengthen our contribution to understanding disempowerment in multi-human environments. We plan to withdraw from this venue and incorporate this valuable feedback in a substantially revised version.

**Withdrawal Confirmation:**

I have read and agree with the venue's withdrawal policy on behalf of myself and my co-authors.